# Opinions of physiotherapists at the University Teaching Hospital on assessing the psychological well-being of patients with stroke: A qualitative study

Taonga Nalungwe[1], Deborah Chileya[1], Joseph Lupenga [2]*

**1** Department of Physiotherapy, Faculty of Health Sciences, Lusaka Apex Medical University, Lusaka, Zambia, **2** Department of Epidemiology and Biostatistics, School of Public Health, University of Zambia, Lusaka, Zambia

* lupengajoseph@gmail.com

## Abstract

Patients with stroke often encounter psychological challenges because of the disabilities and loss of independence resulting from the condition. However, these psychological problems are frequently overlooked or under-reported during stroke rehabilitation by physiotherapists. Hence, this study explored the opinions of physiotherapists at the University Teaching Hospital on assessing the psychological well-being of patients with stroke. A qualitative phenomenological study design was used, and data were collected through in-depth interviews with physiotherapists at the University Teaching Hospital in Lusaka. Ten physiotherapists identified through purposive sampling participated in the in-depth interviews. The analysis was performed using Atlas.ti version 22 software. Thematic analysis was conducted using an inductive approach. Ten semi-structured in-depth interviews with physiotherapists working in the stroke unit identified four themes: psychological assessment as part of stroke care in physiotherapy practice, assessing the psychological well-being of patients after stroke, confidence, and competence in psychological assessment, and professional development needs. The psychological well-being of patients is inconsistently assessed by physiotherapists because of lack of standard guidelines, lack of confidence, lack of competence and inadequate training. This underscores the need to revise the physiotherapy training curriculum, integrate psychological approaches in stroke care, and revise the scope of practice to address current gaps in practice.

## Introduction

Stroke imposes a significant strain on both stroke survivors and the broader community [1]. After experiencing a stroke, individuals often experience motor impairments like muscle weakness, diminished dexterity, and altered sensation [2], which significantly

---

**Data availability statement:** All data underlying the findings described in this manuscript can be accessed through this link https://figshare.com/s/c1e878165a9082b9a16f.

**Funding:** The author(s) received no specific funding for this work.

**Competing interests:** The authors have declared that no competing interests exist.

hamper their engagement in daily activities [3]. Upon discharge from the hospital, stroke survivors commonly experience hemiparesis, challenges in performing routine tasks, limitations in social participation, and cognitive difficulties [4]. Stroke not only carries a high risk of mortality and disability but also serves as a significant precursor to various psychological problems, hindering the recovery process for survivors [5].

Lahey [6] defined psychological problems as "simply aspects of human behaviour that include ways of thinking, perceiving, feeling, and acting that cause people distress or interfere with functioning in important areas of their lives." Psychological problems happen along a continuum from uncomfortable to extremely distressing, problematic, and sometimes tragic [7]. They develop because of the combination of innate genetic predispositions and environmental influences and are universal aspects of human experience instead of distinctive disorders [7]. In patients with stroke, psychological problems are common. For instance, studies indicate substantial rates of anxiety (51.3%) and depression (76.1%) among individuals three months post-stroke [8]. Additionally, post-stroke depression (PSD) affects between 18–33% of individuals, yet remains significantly underdiagnosed and undertreated [9]. Despite being primarily associated with physical health, physiotherapy practice often involves dealing with patients experiencing psychological distress, including depression, anxiety, and suicidal thoughts [10]. Research indicates that these psychological problems, such as PSD, significantly increase the risk of recurrent stroke by 48% [11] and diminish the quality of life [12]. Thus, it is crucial to monitor and address psychological concerns to mitigate their impact on the patient's overall health post-stroke.

The separation of mental and physical health by professionals, institutions, and cultures makes it difficult to provide high-quality care for people post-stroke who experience psychological problems [13]. As a result, services become fragmented, and chances to increase quality and efficiency are frequently missed [13]. Given that patients with stroke frequently present with psychological issues when seeking physiotherapy care, it is critical for physiotherapists to assess their patients' psychological aspects to optimise patient management. According to the findings of a meta-analysis, physiotherapists believe that treating patients who are depressed and anxious is part of their job even though they feel unprepared for it [14]. A narrative review also revealed that physiotherapists lacked the competence and confidence to assess the psychological well-being of patients post-stroke [15].

While past research has indicated that physiotherapists commonly encounter patients with stroke presenting with psychological problems, there remain significant gaps in understanding regarding the readiness of physiotherapists to assess psychological problems in patients with stroke. Thus, this study explored the opinions of physiotherapists at the University Teaching Hospital on assessing the psychological well-being of patients post-stroke.

## Materials and methods

### Ethical statement

All procedures were performed under the Declaration of Helsinki, relevant legislation, and institutional guidelines. The study was approved by the Lusaka Apex Medical

University Bio-Medical Ethics Committee (LAMUBREC) (ref: 00030–23). Both verbal and written consent was obtained from each participant prior to participating in the study. To uphold participant confidentiality, no names were used or referenced in any subsequent documentation.

## Study design

The study utilised a qualitative phenomenological research design to explore the opinions of physiotherapists at the University Teaching Hospital on assessing the psychological well-being of patients post-stroke, to acquire a comprehensive understanding of their lived experiences and their perspective [16,17].

## Study setting and study participants

The study was conducted at the university teaching hospital's physiotherapy department. The University Teaching Hospital (UTH) is the largest government-owned hospital in Lusaka, Zambia. The physiotherapy outpatient unit provides care for outpatients with a wide range of conditions, including stroke. Purposive sampling was used to select study participants. Participants were selected based on their experience in managing patients with stroke [18,19]. The study included physiotherapists with a diploma or higher in physiotherapy who agreed to participate. Those with less than two years of clinical experience, those who were unavailable because of illness or work leave, and those who refused to participate were excluded. A list of available physiotherapists was obtained from the head of physiotherapy department. Eligible participants were approached in person by the author [TN] during departmental working hours. Physiotherapists who met the inclusion criteria were briefed on the study's objectives, participation procedures, and obtaining informed consent. Before their involvement in the study, participants gave both verbal and written consent, indicating their willingness to participate. The sample size was determined based on the principle of data saturation [20,21]. We continued to interview participants until sufficient information had been collected to thoroughly understand the phenomenon under study and no new analytical insights emerged [20,21]. After conducting interviews with ten participants, data saturation was reached. A total of 14 physiotherapists were invited to participate. Ten people agreed to and participated in the interviews, while four others declined, citing a lack of time. No participants dropped out once they consented to participate.

## Data collection

A semi-structured in-depth interview (IDI) guide developed by the authors following the literature review was used to inform data collection. The IDI guide consisted of open-ended questions about the physiotherapists' opinions on the assessment of psychological health in patients post-stoke, the frequency with which they perform the assessments, and factors that could improve these assessments. Before conducting the interviews with the participants, the definition of psychological assessment and psychological problems were provided to the participants. Psychological assessment was defined as the process of gathering data to evaluate a person's behaviour, abilities, and personality through interviews, observation, standardised tests, self-report measures, physiological or psychophysiological measurement devices, or other specialised procedures [22]. Psychological problems were defined as problematic ways of thinking, feeling, and behaving that lie on continuous dimensions and are not necessarily diagnosed mental illnesses [7]. Examples of psychological problems such as depression, anxiety, stress, and PTD were also given to improve their understanding of the topic under discussion. The face-to-face interviews were facilitated by one author (TN) and notes were taken during the interview sessions, while recordings were made using a tape recorder. Throughout the data collection process, ethical guidelines on confidentiality, privacy, and consent were strictly adhered to. The IDIs were scheduled at times and locations convenient to the participants and were conducted in English. Interviews lasted approximately 25–35 minutes. All interviews took place in the staff room of the physiotherapy department at UTH. The data was collected in May and June 2023.

## Trustworthiness

The trustworthiness of this study was ensured by following the four criteria (confirmability, credibility, dependability, and transferability) proposed by Guba in 1981 [23]. To achieve credibility, peer debriefing, a process which involved two authors (TN and DC) was used to maintain credibility by reviewing the analysis and interpretation of the data [23–25]. This ensured that it served as an external check on the research process and identified any biases, assumptions, or points that needed clarification [24]. The notes and observations made during the interviews were also cross-checked and confirmed by the participants themselves [24,26,27]. Detailed explanations of the study research design, data collection procedures, analysis methods, research site, participant demographics, and data collection processes have been provided to ensure transferability [24–26,28]. Dependability has the concept of consistency, whereas confirmability responds to the principle of neutrality [25]. Dependability was achieved by building an audit trail, where the researcher documented every phase of the research process, including decisions made, data collected, and analysis carried out [25,26,29]. Confirmability was achieved by grounding findings in data, avoiding researcher bias. Analysis and interpretation of the findings were not based on the researcher's interests and beliefs but on the data itself [24,25]. Reflexivity was employed, which prompted authors to think about their subjectivity and its influence on data collection and data analysis [24,25]. Interviews were conducted by TN, a physiotherapist who has a bachelor's degree, academic qualitative research training, and over five years of stroke rehabilitation clinical experience. Her background facilitated effective engagement with participants and an understanding of the clinical context. The interviewer had no prior supervisory or personal connections with the participants, which helped minimize power imbalances and possible response bias. The research team also included JL (MSc) and DC (MPH), both with academic backgrounds in physiotherapy and Public Health. All authors were not affiliated with the facility where the study was conducted.

## Data analysis

Analysis was conducted using Atlas.ti version 22 software, using thematic analysis with an inductive approach. Thematic analysis was guided by the six-step process suggested by Braun and Clarke [30,31]: (1) familiarisation with the data, (2) production of initial codes, (3) searching for themes, (4) reviewing themes, (5) defining and naming themes, and (6) writing the research report. Before analysis, the audio recordings were transcribed verbatim and initial notes were made. Data analysis was conducted by one author (JL). The first step involved familiarisation with the interview transcripts to get a general understanding of the data. From each interview transcript, codes were developed to describe the content observed. The data was then organised according to these codes. Initial themes were generated from the codes, sometimes combining several codes to form a coherent theme. To ensure that each theme captured the underlying data accurately and effectively, it was carefully reviewed. Where necessary, adjustments were made to the themes. These included splitting, combining, discarding, or creating new themes to improve their accuracy in describing the data. The themes were later defined, and the wording was refined for better clarity, coherence, and alignment with the study objectives.

## Results

### Demographic information of the participants in the interviews

Ten participants participated in the in-depth interviews. Three men and seven women between the ages of 28 and 51 whose professional experience ranged from 4 to 23 years participated in the study. Three of the participants had a diploma, four had a bachelor's degree and three had a master's degree.

## Themes identified in the in-depth interviews

Four themes were identified, including psychological assessment as part of stroke care in physiotherapy practice, assessing the psychological well-being of patients after stroke, confidence and competence in psychological assessment, and professional development needs (see Fig 1).

**Theme 1: Psychological assessment as part of stroke care in physiotherapy practice.** Participants recognised psychological well-being as an important component of stroke care that is often overlooked. While there was an agreement that the assessment of the psychological status of patients after stroke forms part of comprehensive care and was considered being part of the physiotherapy practice, it is not clearly outlined in the scope of practice for physiotherapists. One participant shared, *"It's not coming out very clearly, but it is an important part of our scope of practice"* (ID 3). Participants emphasised the requirement for psychological assessments, indicating that knowledge of a patient's psychological well-being is just as important as assessing their physical recovery. However, the physiotherapy scope of practice does not clearly integrate psychological assessment into stroke care, and therefore most physiotherapists are uncertain about the standard procedures and expectations for such assessments. Additionally, the current physiotherapy assessment form for patients with stroke only focuses on the patient's physical condition, excluding

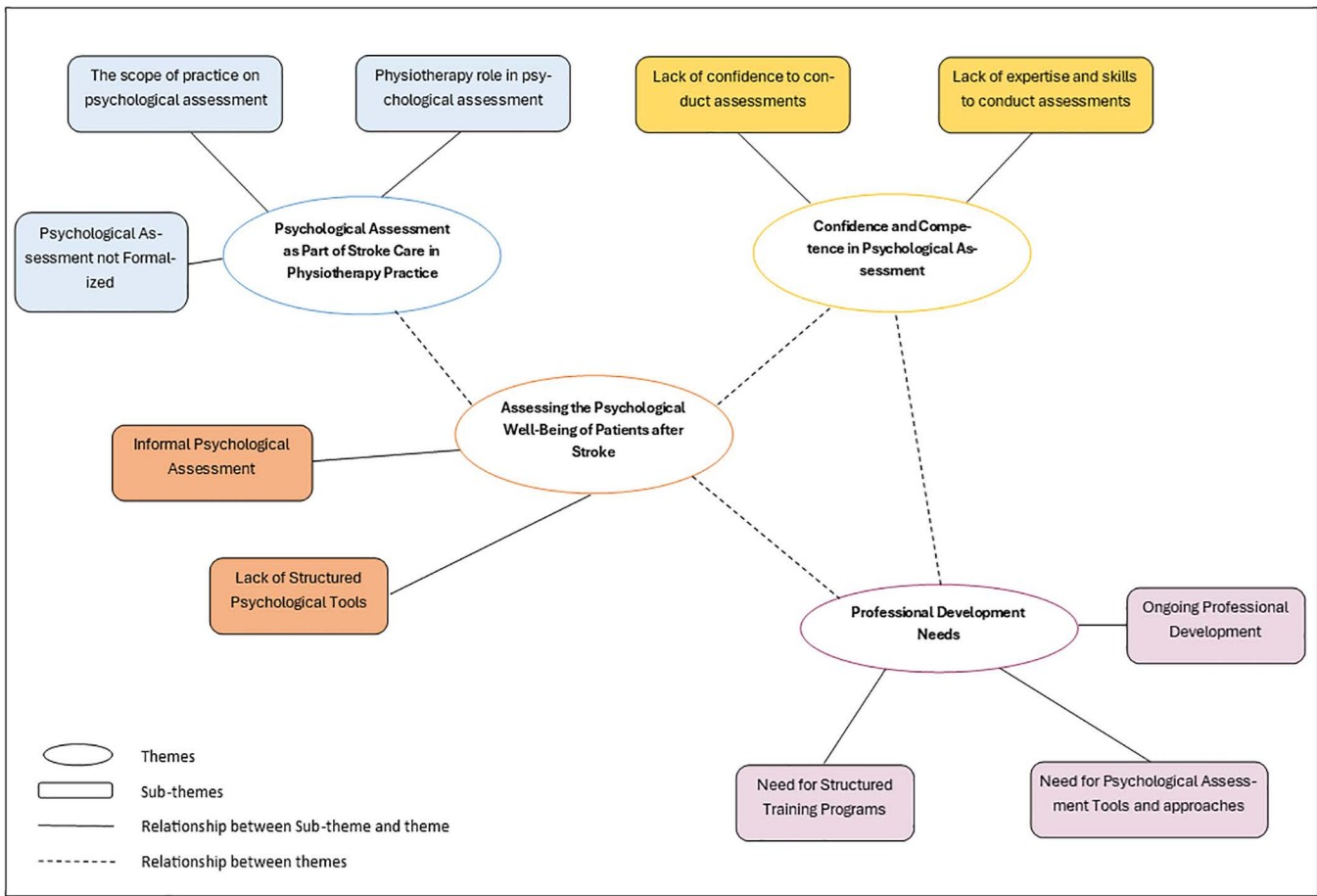

**Fig 1. Mapping of themes in the study.**

psychological well-being. One participant shared, *"No, it's not we I've seen a lot of assessment forms, or you know even when they're talking they're just focusing on their condition…not about their well-being."* (ID 1)

**Theme 2: Assessing the psychological well-being of patients after stroke.**  Participants shared that assessing the psychological well-being of patients with stroke is critical in the management of patients with stroke, as it helps them identify the appropriate physiotherapy treatment approach, communicate with the patient or give instructions to the patient. While participants considered assessing the psychological well-being of patients with stroke to identify psychological problems as one of their responsibilities, others noted a lack of clear guidelines regarding how to perform such assessments. This lack of clear guidelines led to inconsistent practices, where some physiotherapists overlooked the assessment of psychological well-being altogether or only conducted when patients explicitly expressed psychological distress. For some participants, identification occurred only when patients voluntarily expressed or shared their thoughts or feelings with them.

> *"So, I haven't gone into details assessing the mental aspect of stroke. For a number of reasons, one, you know, you have to have a certain tool that you're going to use to assess this or follow a certain protocol. And then this is not something that's really in the scope of physio practice, I would say we learn it, we know we're supposed to do that. And when I see that there's a need for psychological assessment, I would usually refer patients to a psychiatrist, or psychologist."* (ID 5)

For some, the assessment of psychological well-being to identify psychological problems among patients after stroke was based on informal assessment. They emphasised that their informal assessment relied on observing the patient's orientation, responsiveness to questions, commands, or instructions, and general behaviour during the physical assessment, which is not appropriate or adequate for identifying psychological problems.

> *"I ask them to explain something, and I listen to them explain. Yeah, so as I talk to them, I gauge the state of their mind. Because sometimes during an assessment, I give an instruction for the patient to do something. So, I look at how long they take to respond to my command or my instruction, and when they respond, are they responding accurately?"* (ID 3)

**Theme 3: Confidence and competence in psychological assessment.**  Some participants expressed confidence in their ability to assess the psychological well-being of stroke patients. One participant stated, "*Looking at the number of years of practice, I feel qualified*" (ID 6). On the other hand, other participants admitted lacking confidence in the assessment of psychological well-being in patients after stroke, because of a perceived lack of training and expertise in this area. Their confidence in their ability to assess psychological well-being was closely tied to access to resources, such as training in psychosocial counselling and the availability of performance assessment forms. One of the participant shared, "*I am comfortable but not very confident with the assessment, but with specific tools and more training, I can get better*" (ID 7). Many participants also reported lacking the qualifications and expertise needed to assess and manage psychological problems in patients with stroke.

**Theme 4: Professional development needs.**  Professional development was identified as a crucial element in improving the practice of psychological assessment in stroke care. Participants expressed the need for structured training programs that focus on both the psychological assessment of patients with stroke and psychosocial counselling. One participant shared, "*I think it could be a good thing if they were trained*" (ID 2). Workshops and continuing education opportunities were seen as essential for those already in practice, helping them to gain the necessary skills and confidence to integrate psychological assessments into their care routines. Another participant said, *"workshops on how to handle…some of these areas of need for stroke patients like the psychological well-being because it may be difficult sometimes to know, usually actually is difficult for people to know how, how a stroke patient might react…because of their mental state"* (ID 3).

Many participants also expressed the need for psychological assessment screening tools and approaches. They shared that this could help ensure that psychological assessments are thorough, consistent, and documented properly. The screening tools or approaches would provide them with a structured framework or guide on how to approach and assess psychological aspects when assessing patients with stroke. One participant shared, *"I would like to add that there is also a need for the formulation of structured formal psychological assessment tools to guide and enable physiotherapists to confidently assess the psychological well-being of stroke patients"* (ID 9).

## Discussion

This study provides insights into the opinions of physiotherapists at the University Teaching Hospital on assessing the psychological well-being of patients post-stroke. The study revealed that physiotherapists acknowledge the importance of assessing the psychological well-being of patients post-stroke despite not being clearly stated in the physiotherapy scope of practice. The assessment of the psychological well-being of patients post-stroke was inconsistent and was affected by a lack of standard guidelines, a lack of confidence, and competence and inadequate training. Training and adoption of screening tools were cited as crucial elements in improving the practice of psychological assessment in stroke care.

The study revealed that physiotherapists view the assessment of the psychological well-being of patients with stroke as part of their role, despite not clearly stated in their current scope of practice. This finding is supported by the findings from an Australian study which revealed that physiotherapists felt that the assessment of psychological distress was beyond their scope of practice, although they believed that recognising psychological distress was part of their scope of practice [32]. In contrast, a study in Australia highlighted a lack of understanding among physiotherapists regarding their role in the management of individuals with severe psychological problems [33]. These results could be explained because physiotherapists are often trained to focus primarily on physical rehabilitation and place less emphasis on the psychological aspects of patient care. This demonstrates that more work is required within the physiotherapy profession to determine the scope of practice for assessing the psychological well-being of patients undergoing physical rehabilitation. The International Organization of Physical Therapy in Mental Health (IOPTMH) stipulates that all physiotherapists should be able to identify psychological problems by performing assessment utilizing psychological screening tools, clinical observation, and patient interviews), but not diagnose or implement treatment strategies [34]. Integrating psychological assessment within the physiotherapy scope of practice for stroke care is critical considering that psychological distress can affect rehabilitation progression [35]. Identifying psychological problems also allows physiotherapists to collaborate in interdisciplinary settings by referring patients to psychologists [34].

The study reveals inconsistencies in post-stroke patients' psychological well-being assessments; some physiotherapists conduct no assessments, while others rely on informal observations or patient-initiated reports. This confirms the findings of previous studies which reported that physiotherapists rely on informal assessment methods to assess the psychological well-being of patients [36–38]. These inconsistencies highlight the need to strengthen psychological screening within physiotherapy practice to better identify patients at risk of psychological distress. Brief depression screens, integrated suicide/non-suicidal self-harm and depression screens, multidimensional screens and health-related distress measures have been recommended for this purpose [39]. Such screening can facilitate early identification of patients for referral to psychologists for comprehensive assessment, diagnosis and management. The early identification of patients with stroke that could benefit from psychological services offered by psychologists could reduce adverse outcomes like increased mortality risks, heightened stroke susceptibility in post-stroke patients [11,40], and diminished quality of life [12]. The current lack of assessment of psychological well-being to identify patients for referral to psychologists represents a missed opportunity for optimizing recovery and overall health outcomes of patients undergoing rehabilitation.

The study reveals that factors such as lack of standard guidelines, lack of confidence, lack of competence and inadequate training contributed to the inconsistencies in the assessment of psychological well-being. Supporting the findings of this study, previous studies from Australia, Italy, and South Africa also reported a lack of education and limited knowledge

as the barrier to conducting psychological assessment [37,41,42]. This finding corroborates with findings from other studies that cite physiotherapists' lack of competence and skills as a barrier to conducting psychological assessment [35,42,43]. Lack of clinical guidelines is another important factor affecting the assessment of psychological well-being among patients undergoing rehabilitation [35]. These findings underscore the importance of investing in professional development for physiotherapists and integration of psychological approaches in stroke care to build physiotherapists' confidence, competence, and skills in addressing patients' psychological needs during rehabilitation. This also underlines the need to revise the physiotherapy training curriculum to address the current gaps in physiotherapy practice [36]. Participants suggested that physiotherapists in the stroke care unit could benefit from professional development. This aligns with findings from previous studies that emphasized the need for education and training focusing on the clinical presentation of psychological problems, and communication skills to improve healthcare professionals' understanding and assessment of psychological problems in patients [33,35]. Joint training initiatives with psychologists could provide physiotherapists with valuable insights and strategies to address the psychological aspects of patient care. In addition, incorporating psychological approaches to stroke rehabilitation care could be vital, rather than just developing psychological screening tools. Physiotherapy is a profession that is centred more on physical rehabilitation but shifting the focus also to psychosocial factors in clinical practice is crucial to integrating psychological assessment in physiotherapy practice [44].

The study had limitations which include; the study involved ten physiotherapists from a single university teaching hospital. The study did not fully capture the different perspectives and experiences of physiotherapists in different healthcare settings. Physiotherapists at UTH, may have more informed and progressive views on holistic care, including psychological well-being, due to exposure to diverse clinical cases, interdisciplinary collaboration, professional development, and academic engagement. This perspective may differ from physiotherapists in district hospitals, private hospitals, or rural settings with different experiences. Participants may have provided responses that they perceived as socially desirable, leading to potential under-reporting or over-reporting of certain practices or attitudes related to the assessment of psychological well-being.

## Conclusion

There is a lack of assessment of psychological well-being in the assessment of psychological well-being of patients post-stroke with some physiotherapists not conducting any assessments and some relying on informal observations or patient-initiated decisions. Key factors contributing to inconsistencies in the assessment include lack of standard guidelines, lack of confidence, lack of competence and inadequate training. Considering that physiotherapists work with patients with stroke who exhibit psychological problems, they could benefit from training in identifying such problems to facilitate referrals to specialized clinicians for diagnosis and management. Updating the physiotherapy scope of practice for patients with stroke to clearly define the role of physiotherapists in mental health of patients with stroke could help address current gaps and improve patient care. While physiotherapists cannot diagnose or treat, they are responsible for screening patients to identify those requiring further intervention [34]. Integrating the identification of psychological problem through screening, clinical observation, and patient interviews into physiotherapy training curricula is essential to equip physiotherapists with the necessary skills needed to identify patients for referral.

## Acknowledgments

The authors express gratitude to Lusaka Apex Medical University, UTH Hospital Management, and the study participants.

## Author contributions

**Conceptualization:** Taonga Nalungwe, Deborah Chileya, Joseph Lupenga.

**Data curation:** Taonga Nalungwe.

**Formal analysis:** Taonga Nalungwe, Deborah Chileya, Joseph Lupenga.

**Investigation:** Taonga Nalungwe.

**Methodology:** Taonga Nalungwe, Deborah Chileya, Joseph Lupenga.

**Supervision:** Deborah Chileya.

**Writing – original draft:** Joseph Lupenga.

**Writing – review & editing:** Taonga Nalungwe, Deborah Chileya, Joseph Lupenga.

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
