## [Decision Letter · Decision Letter 0]

22 Jul 2025

PMEN-D-25-00138

Opinions of Physiotherapists at the University Teaching Hospital on Assessing the Psychological Well-Being of Patients with Stroke: A Qualitative Study

PLOS Mental Health

Dear Dr. Lupenga,

Thank you for submitting your manuscript to PLOS Mental Health. After careful consideration, we feel that it has merit but does not fully meet PLOS Mental Health’s publication criteria as it currently stands. Therefore, we invite you to submit a revised version of the manuscript that addresses the points raised during the review process.

The manuscript has been evaluated by a reviewer, and their comments are available below.

The reviewer requests more detail on author reflexivity, ethical approval and participant recruitment. They also make suggestions for the results and discussion sections.

Could you please revise the manuscript to carefully address the concerns raised?

Please note that we have only been able to secure a single reviewer to assess your manuscript. We are issuing a decision on your manuscript at this point to prevent further delays in the evaluation of your manuscript. Please be aware that the editor who handles your revised manuscript might find it necessary to invite additional reviewers to assess this work once the revised manuscript is submitted. However, we will aim to proceed on the basis of this single review if possible.

We look forward to receiving your revised manuscript.

Kind regards,

Jenna Scaramanga

Staff Editor

PLOS Mental Health

Journal Requirements:

1.In the online submission form, you indicated that The data transcripts are available from the corresponding author [TN] upon reasonable request.

3. Uploaded as supplementary information.

Reviewers' comments:

Reviewer's Responses to Questions

**Comments to the Author**

1. Does this manuscript meet PLOS Mental Health’s publication criteria?

Reviewer #1: Yes

2. Has the statistical analysis been performed appropriately and rigorously?

Reviewer #1: N/A

3. Have the authors made all data underlying the findings in their manuscript fully available (please refer to the Data Availability Statement at the start of the manuscript PDF file)?

Reviewer #1: Yes

4. Is the manuscript presented in an intelligible fashion and written in standard English?

Reviewer #1: Yes

Reviewer #1: Many thanks to the authors for submitting the manuscript. They address a very relevant topic that should be considered in both the training and professional practice of physiotherapists, providing a holistic approach to patient care after stroke.

The "Trustworthiness" section is particularly important, as it demonstrates the care taken to address some of the study's validity criteria. Although aspects related to the research team and reflexivity are described, it is necessary to present additional elements to better understand the position of the research team, as well as its relationship with the participants. This will enable a contextual understanding of the results and the proposed discussion.

It is recommended that you review the COREQ guide criteria (Domain I) for qualitative studies, available on the EQUATOR network https://www.equator-network.org/

Study participants: How were participants approached? How many people were invited but refused to participate or drop out?

Where was the data collected? At the same hospital?

Ethical Issues: Could you indicate which ethics committee approved the study protocol?

Results: If possible, and if it doesn't compromise the anonymity of the participants, it's probably very useful to describe the participants in a table, indicating the participant number (or ID). This number can be displayed at the end of each quote, making it easier to understand the participants' speech.

Please check that in all textual quotes, the person (participant ID) speaking is indicated.

Discussion: In the limitations section, the authors mentioned that the study involved ten physiotherapists from a single university teaching hospital. Could you expand on this point? How might it differ in other contexts? Perhaps, being a university hospital, they have a broader vision than others?

**Do you want your identity to be public for this peer review?** For information about this choice, including consent withdrawal, please see our Privacy Policy

Reviewer #1: **Yes: ** María Teresa Solís-Soto

---

## [Decision Letter · Decision Letter 1]

5 Oct 2025

PMEN-D-25-00138R1

Opinions of Physiotherapists at the University Teaching Hospital on Assessing the Psychological Well-Being of Patients with Stroke: A Qualitative Study

PLOS Mental Health

Dear Dr. Lupenga,

Thank you for submitting your manuscript to PLOS Mental Health. After careful consideration, we feel that it has merit but does not fully meet PLOS Mental Health’s publication criteria as it currently stands. Therefore, we invite you to submit a revised version of the manuscript that addresses the points raised during the review process.

Your manuscript has been evaluated by two reviewers, and their comments are available below. While Reviewer 1 recommends acceptance, a new reviewer has some suggestions for refinements, particularly to the discussion section. Please carefully revise your manuscript to address the points raised.

We look forward to receiving your revised manuscript.

Kind regards,

Jenna Scaramanga

Staff Editor

PLOS Mental Health

Journal Requirements:

Additional Editor Comments (if provided):

Reviewers' comments:

Reviewer's Responses to Questions

**Comments to the Author**

Reviewer #1: All comments have been addressed

Reviewer #2: (No Response)

publication criteria?

Reviewer #1: Yes

Reviewer #2: Yes

3. Has the statistical analysis been performed appropriately and rigorously?

Reviewer #1: N/A

Reviewer #2: N/A

4. Have the authors made all data underlying the findings in their manuscript fully available (please refer to the Data Availability Statement at the start of the manuscript PDF file)?

Reviewer #1: Yes

Reviewer #2: (No Response)

5. Is the manuscript presented in an intelligible fashion and written in standard English?

Reviewer #1: Yes

Reviewer #2: Yes

Reviewer #1: Many thanks to the authors for submitting a revised version of their manuscript, and incorporate suggestions and recommendations.

There's just one detail: it would be important to standardize all citations, including the ID or participant number (in all citations) in a consistent and organized manner. Perhaps the ID number at the end of the citation would suffice, for example: "....XXX..." (ID 3)

Reviewer #2: This manuscript is well written and the qualitative analyses appear to have been conducted soundly. My main concern revolves around the discussion point that physiotherapists should expand their scope to be trained in psychological assessment. Further details below.

Abstract: clear and concise. Nice job.

Introduction: excellent. clear rationale for the study and nice review of the relevant literature.

Materials and methods: Nicely done overall, good level of detail. Suggested edit: page 6, lines 123-124. The term "rare and terrifying illness of the mind" sounds bizarre to me. If this is a quote from your reference, please use quotation marks. If this is the authors' phrasing, I suggest rewording to something like "...that lie on continuous dimensions and are not necessarily diagnosed mental illnesses."

Results: Table 1 compromises the anonymity of participants. For example, if someone in the physiotherapy department of your institution reads your manuscript, would they not be able to identify who is the 51-year-old male with 23 years of experience? I would recommend removing this table and providing this data in aggregated form (which you do in the first paragraph), rather than specifying the details of every participant individually.

Themes are nicely explained.

Discussion & Conclusion: I recommend that authors reflect more deeply on their discussion point regarding the appropriate scope of physiotherapy (e.g., page 14, lines 289-291 and 296-298; page 16, lines 346-347). Imagine the opposite scenario: psychologists who are assessing patients' mental health should of course consider the impact of physical impairments (e.g., hemiparesis), but is it ethical for them to conduct physiotherapy assessments?

In the discussion, it would be helpful to clarify what authors recommend the extent of this psychological assessment would be. In many countries, psychological assessment is the responsibility of a psychologist with rigorous training in mental health, so this would be beyond the scope of what a physiotherapist would be trained to do. In my opinion, the results point towards a discussion about the benefits of multi-disciplinary care and the usefulness of physiotherapists being trained in identifying the need for a referral to psychology, which the authors have touched upon.

Page 16, line 341: confusing phrasing, some words appear to be mixed up.

The other discussion points and conclusions are well explained and valid.

**Do you want your identity to be public for this peer review?** For information about this choice, including consent withdrawal, please see our Privacy Policy

Reviewer #1: **Yes: ** María Teresa Solís-Soto

Reviewer #2: No

---

## [Editor Report · Decision Letter 2]

15 Oct 2025

Opinions of Physiotherapists at the University Teaching Hospital on Assessing the Psychological Well-Being of Patients with Stroke: A Qualitative Study

PMEN-D-25-00138R2

Dear Mr Lupenga,

We are pleased to inform you that your manuscript 'Opinions of Physiotherapists at the University Teaching Hospital on Assessing the Psychological Well-Being of Patients with Stroke: A Qualitative Study' has been provisionally accepted for publication in PLOS Mental Health.

Best regards,

Jenna Scaramanga

Staff Editor

PLOS Mental Health